



# Depth is Relative: The Importance of Depth on TEP in the Near Surface Environment

Tiera-Brandy Robinson[1], Christian Stolle[1,2], Oliver Wurl[1]

[1]Institute for Chemistry and Biology of the Marine Environment, University of Oldenburg, Wilhelmshaven, Germany
[2]Leibniz-Institute for Baltic Sea Research Warnemünde (IOW), Rostock, Germany

*Correspondence to*: Tiera-Brandy Robinson (tiera-brandy.robinson@uol.de)

**Abstract.** Transparent exopolymer particles (TEP) are a major source for both organic matter (OM) and carbon
transfer in the ocean and into the atmosphere. Consequently, understanding the vertical distribution of TEP and the processes which impact its movement are important in understanding the OM and carbon pools on a larger scale. Additionally, most studies looking at the vertical profile of TEP have focused on large depth scales from 5 to 1000s meters and have omitted the near surface environment. Results from a study of TEP enrichment in the sea surface microlayer (SML) in different regions (tropical, temperate) has shown that while there is a correlation between TEP
abundance and primary production (PP) on larger or seasonal scales, such relationships break down on shorter time and spatial scales. Using a novel small-scale vertical sampler, the vertical distribution of TEP within the uppermost 2 meters was investigated. With a maximum variance of TEP abundance between depths ($1.39 \times 10^6$ µg XG eq$^2$ L$^{-2}$) and a minimum variance of ($6 \times 10^2$ µg XG eq$^2$ L$^{-2}$) the vertical distribution of TEP was found to be both heterogeneous and homogeneous at times. Results from the enrichment of TEP and Chl *a* between different regions
has shown TEP enrichment to be greater in oligotrophic waters, when both Chl *a* and TEP abundance was low, suggesting the importance of abiotic sources for the enrichment of TEP in the SML. However, considering multiple additional parameters that were sampled, it is clear that no single parameter could be used as a proxy for TEP heterogeneity, other probable biochemical drivers of TEP transport are discussed.

## 1 Introduction

The sea surface microlayer (SML), a thin layer 10µm-1mm thick, lays at the top of the ocean. It has distinct chemical, biological and physical properties (Cunliffe et al., 2013; Sieburth, 1983; Wurl et al., 2016) setting it apart from underlaying water (ULW). As the boundary layer between the ocean and atmosphere, it significantly controls the flux of such important parameters as $CO_2$ and organic matter (OM)(Engel et al., 2017; Wurl et al., 2016).

The SML is further characterized by its gelatinous nature (Sieburth 1983), being thoroughly permeated with
extracellular polymeric substances (EPS), the largest faction of which is transparent exopolymer particles (TEP)(Cunliffe and Murrell, 2009; Wurl and Holmes, 2008). These gel particles can form in two ways; abiotically via the collision of colloidal material by physical forces, or biotically via the breakdown and secretion of precursor material from organisms, with phytoplankton being the largest source (Passow, 2002a). These gels are "sticky" by nature and thus can aggregate to themselves but also to other solid particles, making them a large source for the
transport of OM in the ocean (Passow, 2002b). Unattached, TEP have a low density and are positively buoyant (Azetsu-Scott and Passow, 2004), so that unless enough highly dense matter (e.g. mineral, phytoplankton cells, fecal pellets) is attached or a dense enough aggregate is formed to cause sinking, these aggregates will rise to the surface and help to form the SML (Wurl and Holmes, 2008). Meanwhile, when these OM rich aggregates sink, they help to feed both the chemical and biological pumps (Engel, 2004; Mari et al., 2017). Due to the role of TEP in OM and
carbon fluxes both within the ocean and into the atmosphere, it is important to understand what parameters can enhance TEP distribution and enrichment in the ocean. Additionally, because TEP are a part of a complex biochemical process, cross regional examination of TEP can help to understand underlying characteristics of TEP.

There have been multiple studies which have looked at the vertical distribution of TEP in the ocean to understand the rising and sinking of these aggregates and their relation to other parameters (Busch et al., 2017; Cisternas-Novoa
et al., 2015; Kodama et al., 2014; Ortega-Retuerta et al., 2017; Wurl et al., 2011a; Yamada et al., 2017). However,



until recently, most studies have focused on large scale vertical distributions between 5-1000's meters and always considered the top 5-10 meters of the ocean as homogenous. As the importance of the SML in air-sea exchanges has grown, more studies have begun to investigate the relationship and enrichment of the SML in comparison with under laying water (ULW). To date, there is no consistent measuring depth for what is termed (ULW), it is dependent solely on the individual setup of the researchers but is often operationally defined at 1 meter.

The purpose of this study was to investigate the abundance and enrichment of TEP between the SML and ULW, in various regions of the ocean and its relation to biochemical factors. A further aim was to determine if 1 meter depth is a good reference for TEP and other parameters, and how important depth is in sampling within the top 2 meters. We present data from three field campaigns which show the accumulation of TEP in the upper 2 meters and how it relates to water column stratification, primary production and sea surface conditions.

## 2 Methods

### 2.1 Study areas:

Water samples were collected as part of the MarParCloud project Cape Verde campaign in the nearshore water in Cao Vicente, on the research cruise HE491 in the North/Norwegian Sea and fjords and from the research cruise EMB184 in the Baltic sea. The sampling areas represent uniquely different regions; Cao Vicente is oligotrophic tropical water with large influences from Saharan dust deposition, the Norwegian fjords and Baltic Sea are both temperate climates, but the inner and outer Norwegian fjord systems have large interaction from North Atlantic water while the Baltic Sea is semi-enclosed with larger anthropological interaction and little interaction with North Atlantic water.

### 2.2 Sampling: Cape Verde

Samples were taken once a day, weather permitting, between September 18th and October 6th, 2016 within the same nearshore water (~1km) with a total of 12 stations sampled. SML and ULW samples were collected from fisher boats in the nearshore waters. SML samples were collected using the glass plate technique (Cunliffe and Wurl, 2014; Harvey and Burzell, 1972) and ULW was collected from 1 meter depth using a large syringe. Wind speed was recorded using an anemometer placed at the nearby Cape Verde Atmospheric Observatory (CVOA) station. A handheld Global Position System (Garmin etrex) was used to track fisher boat movement during sampling and for coordinates of each sampling station.

### 2.3 Sampling: Norwegian (HE498) and Baltic Sea (EMB184) Research cruises

Norwegian Sea and fjord samples were collected between July 8 to 25th, 2017 aboard the R/V Heinke. Samples were collected once per day, weather permitting, from each station with a total of 13 stations spanning inner fjord, outer fjord and open ocean areas. Samples were collected from both the North Sea and the Norwegian Sea, but for the purpose of clarity will from here on be called the Norwegian Sea. Baltic Sea samples were collected between May 30th and June 10th, 2018 aboard the R/V Elisabeth Mann Borgese with a total of 8 stations used. SML and ULW samples were collected using the radio-controlled Sea Surface Scanner S³ as described in (Ribas-Ribas et al., 2017), which has six rotating glass discs partially immersed in the water to sample the SML by its surface tension. ULW (1 meter depth) and SML water were pumped through two separate flow through systems with onboard sensors at a rate of 1.2L min⁻¹ using peristaltic pumps. SML and ULW water are collected in 1L bottles by the pilot's command and in addition collected into large volume carboys. Large sample volumes (~20L) were collected for multiple analyses by all groups involved in the campaigns. The S³ also collects multiple meteorological parameters; PAR, solar radiation, wind speed, and humidity. Salinity was measured on SML and ULW using a multi-parameter meter (MU 6100 H, VWR) before the collection of the sample into a container, high precision in situ temperature was constantly measured for the SML and ULW using a reference thermometer (P795, Dostmann Electronics GmbH). Specifications for instrument precision and accuracy can be found in Ribas-Ribas et al. 2017. All in situ data was averaged for the 2 hours surrounding the sampling of discrete water samples. A new device termed the "High-volume Sampler for the Vertical (HSV)" was deployed to collect water from five depths between the SML and 2



meters. The HSV is made of a vertical polypropylene pipe with five polypropylene tubes set at five distinct depths in the pipe and a float attached to the top which has been ballasted to ensure accuracy in depth. Peristaltic pumps, similar to that on the $S^3$, pump water into collection containers. The HSV was deployed during the collection time of discrete SML and ULW samples by the $S^3$ and near enough to the $S^3$ so that it would sample the same body of water but wouldn't interfere with the glass plate sampling.

**2.4 POC, PON, POP and Nutrients**

Samples for particulate organic carbon (POC), nitrogen (PON) and phosphorous (POP) were filtered onto acid-washed and pre-combusted glass-fiber filters (Whatman GF/C). Filters for POC and PON were dried at 60°C for three days (Norwegian cruise) or 130°C for two hours (Baltic cruise and Cape Verde), put in tin capsules and measured using an elemental analyser (Thermo, Flash EA 1112 and Elementar Analysensysteme, precision of 0.01 ±0.2 ‰). POP was measured by molybdate reaction after digestion with potassium peroxydisulfate ($K_2S_2O_8$) solution (Wetzel and Likens, 2000). The filtered water was collected and analysed for dissolved nutrients ($PO_4$, $NO_3$) by a continuous-flow analyser according to (Grasshoff et al., 1999).

**2.5 Chlorophyll a**

During the Cape Verde and Baltic (EMB184) campaign, Chlorophyll a (Chl *a*) was measured by filtering 500-1000ml of seawater onto pre-combusted (4h, 450°C) GF/F filters (Whatman). The filters were stored frozen (-18°C) until processed. Chl *a* was then analysed according to the method described by (Wasmund et al., 2006) using a fluorometer (Jenway 6285, precision of 0.01 ± <1ng/ml ). During the Norwegian cruise (HE491), in vivo Chl *a* was measured with a hand fluorometer (TURNER DESIGNS, AquaFluorTM, precision of 0.001 absorption) and related to ug of Chl *a* using calibration factor between filtered Chl *a* (Chl *a* Standard in EtOH as reference) and in vivo absorbance.

**2.6 Bacterial cell numbers (only for Baltic cruise)**

For determination of bacterial cell numbers, water samples were fixed with glutaraldehyde (1% final concentration), incubated at room temperature for 1 h, and stored at –18°C until further analysis. Prokaryotic cells were stained with SYBR Green I (2.5 mM final concentration, Molecular Probes, Schwerte, Germany) for 30min in the dark. Samples were measured on a flow cytometer (C6 FlowCytometer, BD Bioscience, fluorescence accuracy of FITC <75; PE <50), and cells were counted according to side-scattered light and emitted green fluorescence. We used 1.0 µm beads (Fluoresbrite Multifluorescent, Polysciences) as internal reference to monitor the performance of the device.

**2.7 Transparent Exopolymer Particles (TEP)**

TEP was measured by filtering seawater onto 0.2um polycarbonate filters under low vacuum (<100mm Hg) and staining with alcian blue solution (0.02 g alcian blue in 100 mL of acetic acid solution of pH 2.5) for 5 sec. 0.2µm filters collect both large TEP aggregates and smaller colloidal TEP material. Filters were stored at -18°C until processed. Alcian blue stain was extracted for 2 hours in 80% Sulfuric Acid, with gentle agitation applied to reduce bubble formation, and analysed using a spectrophotometer (VWR UV-1600PC, precision of  1 ± 0.2% T) and the spectrophotometric method (Passow and Alldredge, 1995). The stock solution of alcian blue was calibrated using xanthan gum (Carl Roth) standard according to (Passow and Alldredge, 1995) . TEP concentrations are shown in relation to xanthan gum equivalence. Recent calibration issues with xanthan gum were not observed in our studies and thus the new method by (Bittar et al., 2018) was not required.

**2.8 Primary production**

To estimate local primary production, we used an adjusted version of the *Vertically Generalized Production Model* (VGPM) (Behrenfeld and Falkowski, 1997) as described by (Wurl et al., 2011b). Estimation is based on concentration of Chl *a*, depth of euphotic zone estimated from the Secchi depth, photoperiod and photosynthetic active radiation (PAR)

**2.9 Data analysis**



Statistical analyses of the data set were performed using Graphpad PRISM Version 8. Differences, null hypothesis testing, and correlation were considered significant when p < 0.05. The data were log transformed, if required, for parametric and analysis of variance (ANOVA) tests, further post hoc tukey analysis was run for comparison of means when ANOVA was significant. Unless otherwise indicated, results are presented as means ±standard
deviations. Enrichment factors (EF) were calculated as the ratio of concentrations in the SML sample to that of the corresponding subsurface bulk water sample. For vertical sample profiles, the variance of each depth measurement from the average was used to determine homogeneity. Variance is the squared deviation from the mean of all depths and is thus given in units squared (e.g. $ug\,^2L^{-2}$).

## 3 Results

### 3.1 General Conditions

General characteristic of parameters for all three campaigns is shown in Table 1 and enrichment in Table 2. We observed low ($<2\,m\,s^{-1}$), moderate ($2-5\,m\,s^{-1}$) and high ($>5\,m\,s^{-1}$) wind regimes (Wurl et al., 2011b). Average wind speed was $3.8\pm0.3$, $4.2\pm2$, $5.6\pm1.8\,m\,s^{-1}$ for the Baltic Sea, Norwegian Sea, and Cape Verde respectively. PAR
averages were $1172\pm145$ and $739\pm251\,µmol\,m^{-2}s^{-1}$ for the Baltic sea and Norwegian sea respectively and sea surface temperature (SST), measured from the SML, was $14.8\pm1.9$ and $14.9\pm1.4°C$. Stations for the Baltic cruise were sampled within the same area (~1km) and thus had similar salinity measurements ($8.92\pm0.2$ psu) relative to the Norwegian cruise. Meanwhile, the stations for the Norwegian cruise covered inner and outer fjord and open ocean areas and thus had larger differences of salinity. SML salinity was ($32.4\pm2$ psu; $23.5\pm0.4$ psu; $6.7\pm3.5$ psu) for outer
fjord/open ocean, Trondheim fjord and Sognefjord, respectively. ULW salinity was ($32.7\pm2$ psu; $23.9\pm0.2$ psu; $6.6\pm3.5$ psu) for outer fjord/open ocean, Trondheim fjord and Sognefjord, respectively. PAR, SST and salinity data were not collected for the Cape Verde campaign due to logistical constraints. Primary Production (PP) ranged from $426-734\,mg\,m^{-2}\,d^{-1}$ during the Baltic cruise but had a higher range during the Norwegian cruise with $318-1194\,mg\,m^{-2}\,d^{-1}$, again this is likely due to the differing water masses sampled during the Norwegian cruise.

### 3.2 TEP distribution in the SML across different regions:

**Baltic Sea**

TEP concentrations ranged from $123-1340\,µg\,XG\,eq\,L^{-1}$ in the Baltic Sea. Nitrate and Phosphate were at the lower limit of detection (nitrate: $<0.1\,µmol\,L^{-1}$; phosphate: $<0.2\,µmol\,L^{-1}$), however Silicate was relatively high with an average of $14.23\pm0.9\,µmol\,L^{-1}$. The Baltic sea was also marked with the highest levels of POC in the SML with a
range of $27.4-274\,µmol\,L^{-1}$. POC enrichment in the SML matched TEP and PON enrichment trends which showed enrichment factors (EF)>1 for St. 4, 5 and EF<1 for St. 9,10. TEP enrichment was ≥1 factor for the beginning of the cruise (St.3-5) and <1 factor for the second half of the cruise (St.8-12). However, total TEP concentration in the SML and ULW increased substantially in the second half of the cruise (St. 9-12), with TEP in the SML averaging $341\pm150$ and in the ULW $269\pm104\,µg\,XG\,eq\,L^{-1}$ at the beginning and in the SML $946\pm386$  and in the ULW
$1916\pm671\,µg\,XG\,eq\,L^{-1}$ for the second half. Chl *a* wasn't enriched in the SML at any station while Chl *a* concentrations ranged between $0.68-1.56\,µg\,L^{-1}$ with the highest concentrations at St. 9 and 10. PP matched trends of TEP except at St. 11, which showed relatively low levels of Chl *a* ($0.80\,µg\,L^{-1}$) and a resulting decrease in PP (from 734 down to 553 $mg^{-2}\,d^{-1}$) but relatively high levels of TEP ($2313\,µg\,XG\,eq\,L^{-1}$) (Fig. 2b).

**Norwegian sea**

TEP concentrations during the Norwegian cruise ranged from $50-424\,µg\,XG\,eq\,L^{-1}$  and had sporadic enrichment with 50% of observations showing EF≥1 and 50% showing EF<1. The highest enrichments were observed at St. 3 (EF=1.6) which was the furthest open ocean station and St. 13 and 14 (EF=1.5; 1.4) which were in the Trondheim fjord. Nitrate and Phosphate were both homogenously low for all stations ($0.04\pm0.04$; $0.07\pm0.03\,µmol\,L^{-1}$) while silicate had a larger range of $0.26-7.98\,µmol\,L^{-1}$ with all outer fjord stations and open ocean stations showing low
($<1\,µmol\,L^{-1}$) concentrations and both inner fjords showing higher ($>1\,µmol\,L^{-1}$) concentrations. The Sognefjord had markedly higher silicate concentrations than the Trondheim fjord with an average of $7.62\pm0.6\,µmol\,L^{-1}$ versus



1.25 and 2.12 µmol L$^{-1}$. PON concentrations in the SML ranged 0.6-2 µmol L$^{-1}$ and were never enriched, mainly due to low over all concentrations in the water. However, PON in the ULW was higher (>1 µmol L$^{-1}$) in both inner fjords compared to the outer fjord and open ocean (<1 µmol L$^{-1}$). POC in the ULW was also higher in both inner fjords (20.5±6.1 µmol L$^{-1}$) compared to the outer fjord and open ocean stations (10.7±1.1 µmol L$^{-1}$).similar to PON, POC in the SML showed no general enrichment and had EF<1 for most stations except St. 3, 8, 11. Chl $a$ concentrations in the SML ranged from 0.29- 1.64 µg L$^{-1}$ with lowest concentrations in the outer fjords and open ocean stations and highest concentrations in the Trondheim fjord. Enrichment of TEP and Chl $a$ were both sporadic and did not have matching trends, with Chl $a$ sometimes enriched when TEP wasn't (St. 5, 12) and TEP enriched when Chl $a$ wasn't (St.14,15). However, this appears to be influenced by the fjord systems, when only the open ocean and nearshore stations were considered, TEP and Chl $a$ enrichment trends did match.

**Cape Verde**

The nearshore water in Cao Vicente, Cape Verde is oligotrophic, which was supported by low Chl $a$ concentrations during our campaign (SML: 0.28±0.2 µg L$^{-1}$; ULW: 0.29±0.1 µg L$^{-1}$). Enrichment of Chl $a$ in the SML was sporadic, with 4 out of the 12 stations showing EF>1 and 5 out of the 12 stations showing EF<1. TEP concentrations in the SML ranged from (94- 187 µg XG eq L$^{-1}$) and were enriched (EF>1) for all days except day 4 and 12. Enrichment of TEP began high at the start of the campaign with (EF=2.6) and was relatively high for the first five days and then decreased to just above unity for the last half of the campaign, excluding the two days of depletion previously mentioned. Of the three regions, samples from Cape Verde showed the lowest TEP concentrations, however not to the same order of magnitude that Chl $a$ showed. Phosphate levels were similar to the other regions with (0.09±0.1 µmol L$^{-1}$) but nitrate levels were higher (0.37±1.3 µmol L$^{-1}$). Silicate concentrations in the SML were lower than in the other regions with an average of 0.95±0.4 µmol L$^{-1}$. Silicate SML concentration and EF had the same trend as TEP EF with higher enrichment at the beginning of the campaign, which then begin to drop in the second half. POC and PON data ranged from 37±32.1 and 2±0.2 µmol L$^{-1}$ with higher values at the first half and lower values in the second half of the campaign. Unfortunately, POC and PON data are only available for half of the stations but are temporally spaced well to assist in showing trends.

### 3.3 TEP, Chl $a$ and POC in different regions

A Tukey's one way analysis of variance was used to compare the concentration and enrichment of the main three parameters between each region: TEP, Chl $a$ and POC. Table 2 shows that TEP concentrations were significantly higher in the Baltic Sea and significantly lower in Cape Verde (SML: p<0.0005, n=11; ULW: p<0.0009, n=11) while TEP enrichment was significantly higher in Cape Verde and significantly lower in the Baltic Sea (p<0.0418, n=8). Samples from the Norwegian cruise fell between the other two cruises in significance for all parameters. Chl $a$ concentrations matched TEP with significantly higher concentrations in the Baltic Sea and lower in Cape Verde (SML and ULW: p<0.0001, n=8). However, Chl $a$ enrichment had no significant difference found between the regions, likely due to overall low enrichment values. POC enrichment was significantly lower in the Norwegian Sea than in Cape Verde (p<0.0062, n=5) but not compared to the Baltic Sea. Statistical analysis for POC could not be run using data from the Baltic Sea due to low n value (n=4). However, both SML and ULW POC concentrations were not significantly different between the Norwegian Sea and Cape Verde but POC enrichment was (p<0.01, n=6).

### 3.4 Vertical Distribution of TEP

The vertical distribution of TEP is shown in Table 3. Variance between concentrations is used to express the relative homogeneity of the parameter within the upper 2 meters and results are given in units squared. During the Baltic cruise there was a distinct change in TEP distribution between the first and second half of the cruise. TEP concentrations were lower and homogenous (average variance =8.63x10$^3$ µg XG eq$^2$ L$^{-2}$) for stations 3-8 but rose in concentration and became heterogeneous (average variance =6.47x10$^5$ µg XG eq$^2$ L$^{-2}$) during stations 9-12. Variances of Chl $a$ were highest at stations 9,10 and 12 and lowest at stations 3-5, showing a positive linear correlation between average vertical concentration and homogeneity (R$^2$=0.95, p<0.0001, n=8): no such correlation was observed for TEP (R$^2$=0.29, p=0.16, n=8). The vertical profiles for microbial counts were also taken in the Baltic Sea to investigate if there was any correlation to TEP depth profiles due to the importance of the microbial





230   loop in TEP production and consumption(Busch et al., 2017; Yamada et al., 2013). However, no correlation or direct
      connection could be found between TEP profiles and microbial profiles, given as total cell numbers (TCN) and
      small autotroph profiles (supplementary material: Figure S1). This is possibly due to the short depth scale or the
      region, Yamada et al. 2017 found a positive correlation in the Arctic Ocean but this was on a much larger depth
      scale (0-4000 meteres) and no correlation was found in the Pacific Ocean. Additionally, any *in situ* production of
235   TEP or consumption by prokaryotes was likely masked by the large increase in phytoplankton abundance during the
      second half of the cruise.

      During the Norwegian cruise, the vertical distribution of TEP varied greatly between stations with the highest
      variance at station 3 (open ocean station) and the lowest variance at stations 5,7 and 11 (fjord/nearshore). There was
      no relation between TEP variance and geographical location, e.g. near shore vs. fjord systems, vs. open ocean.
240   Additionally, no correlation was found between TEP and turbulent kinetic energy (TKE). TKE data during the
      Norwegian cruise are presented in Banko-Kubis et. al. 2018. Thus, TEP vertical distribution is shown based solely
      on its variance (Fig 4) to show how much it can change within the Norwegian sea. It is important to note that station
      3 had a variance nearly 24 magnitude times larger than the second highest variance (st. 9). Chl *a* and POC showed a
      moderate correlation between concentration and variance (Chl *a*: $R^2$=0.67, p<0.0006, n=13; POC: $R^2$=0.63,
p<0.0013, n=13). However, TEP showed no similar correlation when the putative outlier variance from St. 3 was
      excluded ($R^2$=0.02, p=0.66, n=12). During both cruises, TEP was found to be enriched even when POC wasn't, but
      POC was never enriched without TEP also being enriched.

      **4 Discussion**

TEP is one of the main drivers for the transformation of DOM to POM and its uptake into the biological pump.
      Thus, it is important to understand the vertical distribution of TEP and what parameters drive its distribution.
      Previous studies focusing on vertical TEP distributions have considered depth on large scales of 5-1000 meters
      (Cisternas-Novoa et al., 2015; Kodama et al., 2014; Ortega-Retuerta et al., 2017) and has been found to vary greatly
      depending on depth. Due to operational interference from research vessels and the use of large rosette water
samples, most studies sample at 3-5 meters for the shallowest depth and assume this surface water to be
      homogenous towards the surface, and therefore equally representative. However, the importance and influence of
      the SML has been thoroughly supported (Cunliffe et al., 2013; Engel et al., 2017; Hardy, 1982; Liss and Duce, 1997;
      Wurl et al., 2011b) and thus there is a need to better understand the biogeochemical cycling occurring in the near
      surface water and how they relate to organic matter transfer to deeper water masses. This study is the first to take a
higher resolution look at the vertical distribution of TEP and other related parameters in the upper 2 meters of the
      ocean. Our results show that the variability of multiple parameters can be high within the near surface water, due to
      a complex biochemical system, and can occur on much smaller depth scales than previously assumed.

      **4.1 Relation between Chl *a* and TEP enrichment**

Comparing the enrichment of the SML between each region showed a higher variability of enrichment within each
      region than between the regions (Table 2), supporting that SML enrichments is a global phenomenon (Wurl et al.,
      2011b). Interestingly, while there was a significant but weak correlation between Chl *a* and TEP abundance in both
      the ULW and SML (ULW: $R^2$=0.32, p<0.0007, n=30; SML: $R^2$=0.36, p<0.0005, n=30), there was no significant
      correlation between the enrichment of TEP and enrichment of Chl *a* ($R^2$=0.045, p=0.27, n=30). Further supporting
that while phytoplankton are the main source for TEP production, the transport mechanisms for TEP and
      phytoplankton differ. In large part due to the motility of phytoplankton species, which are known to have vertical
      migration patterns (Bollens et al., 2010; Schuech and Menden-Deuer, 2014) and can have motility responses to
      physical changes like turbulence (Sengupta et al., 2017).

      While the highest abundances of TEP were found in the Baltic sea, and the lowest abundances in Cape Verde (Fig
1d), the highest enrichment factors were found in Cape Verde (Fig 1c). As the near shore Cape Verde waters are



oligotrophic, this compliments with previous studies by (Wurl et al., 2011b) which found the highest enrichment of surfactants to be in oligotrophic waters compared to mesotrophic and eutrophic.

### 4.2 Effect of wind Speed on TEP Enrichment

We observed enrichment of all parameters irrespective of either instantaneous wind speeds (2 hour average) or wind speed history (24 hour average), including higher wind speeds > 7 m s$^{-1}$. This supports previous studies which found enrichment of material even at wind speeds >8 m s$^{-1}$ (Kuznetsova et al., 2004; Reinthaler et al., 2008), including the enrichment of TEP in the SML at moderate wind speeds (Wurl et al., 2009). Breaking waves from moderate wind regimes can create bubble plumes in the near surface water (Blanchard and Woodcock, 1957; Deane and Stokes, 2002), and this bubbling has proven to be an effective transport mechanism for TEP and DOM (Robinson et al.,
2019; Zhou et al., 1998) to the SML. Thus, bubbling and turbulence at moderate wind speeds can induce more complex enrichment processes and subdue any direct correlation with wind speed. We never observed wind speeds greater than 8 m s$^{-1}$, which has been found to be the threshold speed for the breakup of TEP (Sun et al., 2017) during experiments in a wind-wave tunnel. Thus, the moderate wind speeds we observed likely had an indirect positive effect on enrichment.

### 4.3 Effect of PP on TEP Enrichment

While TEP concentrations mimic Chl $a$ or PP due to the large contribution phytoplankton play in TEP creation (Ortega-Retuerta et al., 2017; Passow, 2002a), the enrichment of TEP is driven by many other processes. (Wurl et al., 2011a) found TEP enrichment to be irrespective of PP or negatively related with highest enrichment in oligotrophic waters with the lowest PP. Considering the relationship of PP and TEP enrichment within each region,
we found this to be true for the Baltic Sea but not for the Norwegian Sea. In the Baltic Sea, as PP increased enrichment of TEP decreased, due to a larger increase of TEP concentration in the ULW caused from a post bloom state (Fig. 2). However, in the Norwegian sea, TEP enrichment matched PP, most likely due to the changing water bodies in that study, whereas the same body of water was sampled over time for the Baltic Sea and offshore Cape Verde. While we do not have PP data for Cape Verde, considering the positive relationship between PP and Chl $a$
concentration in the SML for the Baltic and Norway data, Chl $a$ concentration in the SML can be used here as proxy for PP in Cape Verde. Under this premise, Chl $a$ in Cape Verde SML was similar to the Baltic in that while, Chl $a$ and TEP SML concentrations were correlated ($R^2$=0.68, $p$<0.012, n=8), enrichment wasn't ($R^2$=0.19, $p$=0.24, n=8).

The oligotrophic waters of Cape Verde present an interesting scenario for TEP production. We found the highest enrichment of TEP in the SML here, while simultaneously observing the lowest concentrations of both Chl $a$ and
TEP. We suggest that this is due to the abundance of precursor material as well as the increased formation of TEP via the abiotic pathway. A tank experiment with the same setup as used by (Robinson et al., 2019), with different techniques to bubble the water, was also employed in Cape Verde, using water from the same near shore area as field samples. The unfiltered water was bubbled using the waterfall technique (Cipriano and Blanchard, 1981; Haines and Johnson, 1995) and via bubbling, large abundances of TEP were created (supplementary Figure S2).
This suggests that while TEP abundance in the near shore water was low, the colloidal and precursor material for TEP was present and only required sufficient formation mechanisms to form aggregates in the size range to be identified as TEP. Such pre-cursor material may have been deposited from the atmosphere, Cape Verde is known for its Saharan dust deposition events and indeed dust events were observed during our campaign (supporting data will be shown in this special issue). This dust deposition has been shown to increase the abiotic formation of TEP (Louis
et al., 2017) and is potentially a large contributing factor to higher enrichment of TEP in Cape Verde. If true, this presents interesting implications for the residence time of dust which ends up floating in the SML with sufficient time for photochemical processing.

In contrast, in mesotrophic and eutrophic water, there is more biological activity present in the ULW which can increase the complexity of the system by which TEP and its precursor material is recycled or altered before it can
reach the SML. Such increase in biologically derived complexity can be seen in the depth profile data from the Baltic Sea, which showed increased heterogeneous mixing of TEP in the water when PP was higher during the second half. Indeed, the HSV data for all parameters shows that the vertical flux processes are not straightforward to interpret.





### 4.4 Down and upward flux of TEP

Previous studies have found chemical characteristics to be heterogeneous in the upper 1-2 meters (Goering and Wallen, 1967; Manzi et al., 1977; Momzikoff et al., 2004) substantiating the notion that vertical flux processes are too complex and strong to assume homogenous mixing. Additionally, phytoplankton communities and abundance have also been found to be heterogeneous in the water column (Cheriton et al., 2009; Dekshenieks et al., 2001; Mitchell et al., 2008). Considering the importance of ULW concentrations in estimating enrichment, the

presumption of ULW as homogenous becomes problematic. When considering a parameter like TEP which bridges the boundary of biological and chemical parameters and is so fundamentally affected by both, these studies become even more crucial indicators for the likelihood of TEP distributions to be heterogeneous, at least in surface water.

Vertical profiles were sampled for the Baltic and Norwegian seas and in both regions, the vertical distribution of TEP, Chl *a*, POC, PON were found to change daily. In the Baltic Sea, the vertical variance of TEP appears to be

linked to PP and the creation of TEP from the biotic pathway. Higher deviation was seen in the second half when TEP and Chl *a* abundance increased. One possible biological cause for this heterogeneous mixing could come from its link to phytoplankton. For example, (Bjørnsen and Nielsen, 1991; Carpenter et al., 1995; Nielsen et al., 1990) found vertical phytoplankton patches within the water column in open North Sea and Baltic Sea. Additionally, (Cheriton et al., 2009) found that vertical oscillations cause stratification of phytoplankton into thin vertical patches.

Thus, if these same processes were to occur in the near surface, stratification of plankton could result in TEP precursor material being released in patches, and with rapid aggregation, create heterogeneous distribution of TEP. Especially with the majority of TEP produced by diatoms, which are non-mobile phytoplankton who would be more susceptible to grouping and patchiness by physical forcing. This is hinted at by the heterogeneous mixing of Chl *a* observed during both cruises (Fig 3 and 4), which always showed heterogeneous mixing. However, this cannot be

the only mechanism as the peak TEP abundance was not always seen at the same depth as Chl *a*. Further studies on vertical phytoplankton distribution in the near surface (>2 meter) environment are needed in order to substantiate the role their patching might have on DOM and TEP. Furthermore, there was a significant correlation between high variance of Chl *a* and high variance of TEP in the Baltic Sea ($R^2$=0.65, p<0.028, n=7, st. 4 excluded) but no correlation in the Norwegian Sea ($R^2$=0.00, p=0.92, n=13), suggesting that with sufficient phytoplankton

abundances, and reduced influence from the open ocean, the biological influences on TEP heterogeneity can dominate.

While biological sources are likely to determine the chemical characteristic of TEP, they aren't the only influence. TEP are operationally defined polymers of acidic polysaccharides, and naturally positively buoyant with highly surface active properties (Zhou et al., 1998). When unballasted by detritus or other organic matter, TEP have a

positive buoyance and can rise to the surface at rates of 0.1-1 m d$^{-1}$ (Azetsu-Scott and Passow, 2004). However, TEP is never unattached from some type of OM and it is this OM which helps to determine the sinking velocity of TEP (Passow, 2002a). Additionally, the density of TEP is dependent on the formation of its precursor material e.g the resulting density and therefore sinking or rising velocity of TEP produced from diatoms vs. bacteria will differ as well as TEP produced from nutrients depletion vs. temperature stress (Mari et al., 2017). In near surface water where

the ambient density of water is stratified, this could result in the immobilization of TEP particles into layers of water with equal density. Thus, heterogeneous mixing of TEP may be caused by a sort of vertical filtering via TEP density and surrounding water density.

### 5 Conclusion

The vertical profiles for TEP, Chl *a* and POC during the Norwegian cruise showed no correlation with any of the sea state parameters. The same was true during the Baltic cruise, which had matching increases in TEP and Chl *a* concentrations but differing depth profiles (Fig 3). On seasonal scales, TEP has been shown to match Chl *a* and POC trends (Mari et al., 2017; Ortega-Retuerta et al., 2017; Wurl et al., 2011a; Zamanillo et al., 2019) supporting the notion of phytoplankton blooms as a main source for TEP production in the ocean and subsequently TEP as a main

source of POC uptake. This is corroborated in our data. However, when considering the vertical transport of these parameters, this relationship is broken or interrupted by the influence of additional mechanisms. Due to the lack of





direct correlation between any one parameter and TEP abundance or enrichment, we conclude that the vertical flux mechanisms of TEP in the near surface environment are complex. Any positive effects on enrichment, such as wind speed and bubble formation, are only partially responsible. However, with the consistent changing of vertical

profiles of TEP, it is clear that these complex fluxes can often result in heterogeneous layering of TEP within the upper 2 meters of the ocean. Indeed within a few cm, TEP abundance can change by up to 291%, with no parameter acting as a proxy to suggest homogeneity or heterogeneity. Therefore, it is important for future studies to accommodate this uncertainty of ULW values and for a standardised depth for all ULW to be incorporated.

**Data availability**

Data have been submitted to PANGAEA database and a doi will be supplied once it is received.

**Author Contribution**

All authors have substantially contributed to this manuscript by their involvement in the research cruises, the collection of data, analysis of the data, or contributing to the writing and discussion of the manuscript.

**Competing interests**

The authors declare that they have no conflict of interest.

**Acknowledgements**

This study received funding from the Leibniz-Society (MarParCloud, SAW-2016_TROPOS-2). We greatly appreciate technical assistance by the crews from both the R/V Heinke and R/V Elisabeth Mann Borgese during the research cruises. Further we would like to thank the MarParCloud group for collaborative efforts during the Cape

Verde campaign, especially Manuela van Pinxteren, Nadja Triesch and Khanneh Wadinga Fomba for coordination efforts. From the University of Oldenburg, we would like to thank Maren Striebel and Hanna Banko-Kubis for help sampling during the Norwegian cruise and Heike Rickels for nutrient analysis.



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



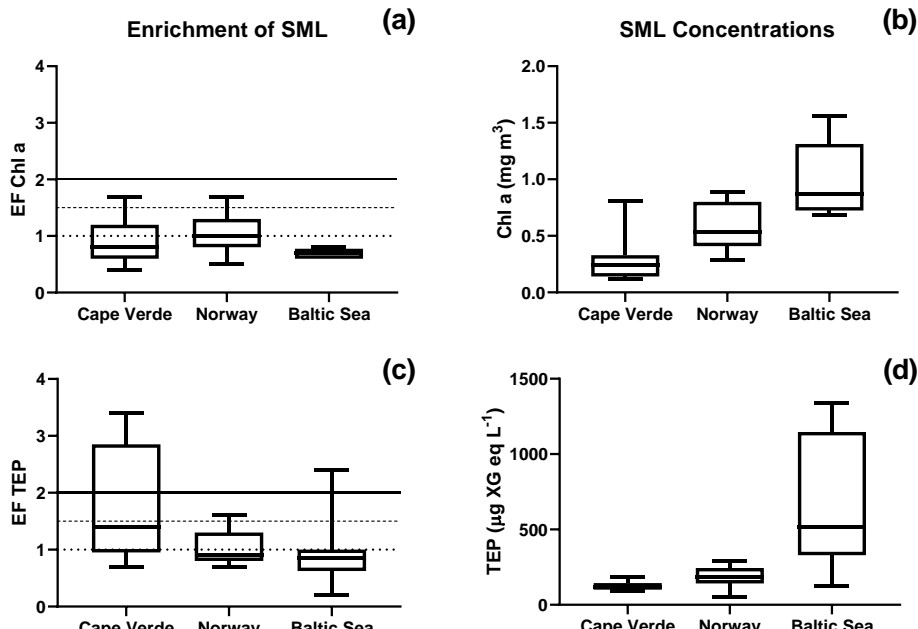

**Figure 1:** Regional comparison of enrichment and SML concentrations for Chl *a* (**a,b**) and TEP (**c,d**).




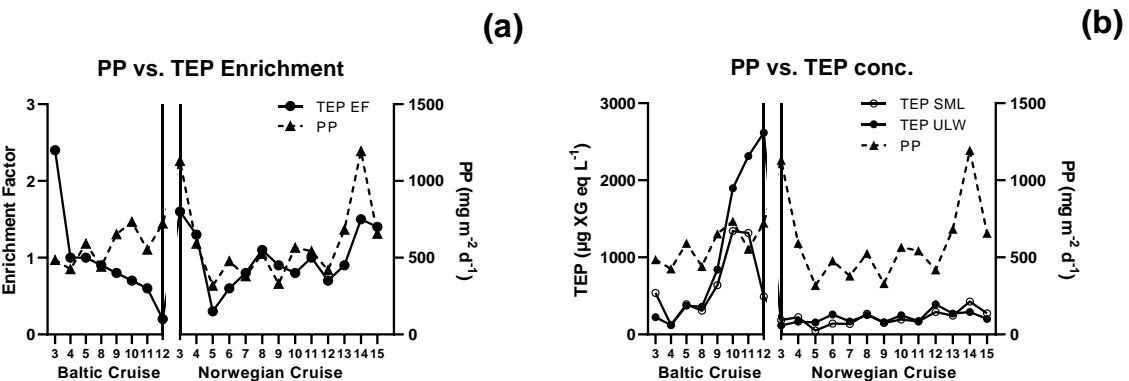

**Figure 2:** Comparison of **(a)** TEP enrichment and **(b)** concentration with primary production (PP) along the cruise tracks for the Baltic and Norwegian cruises.




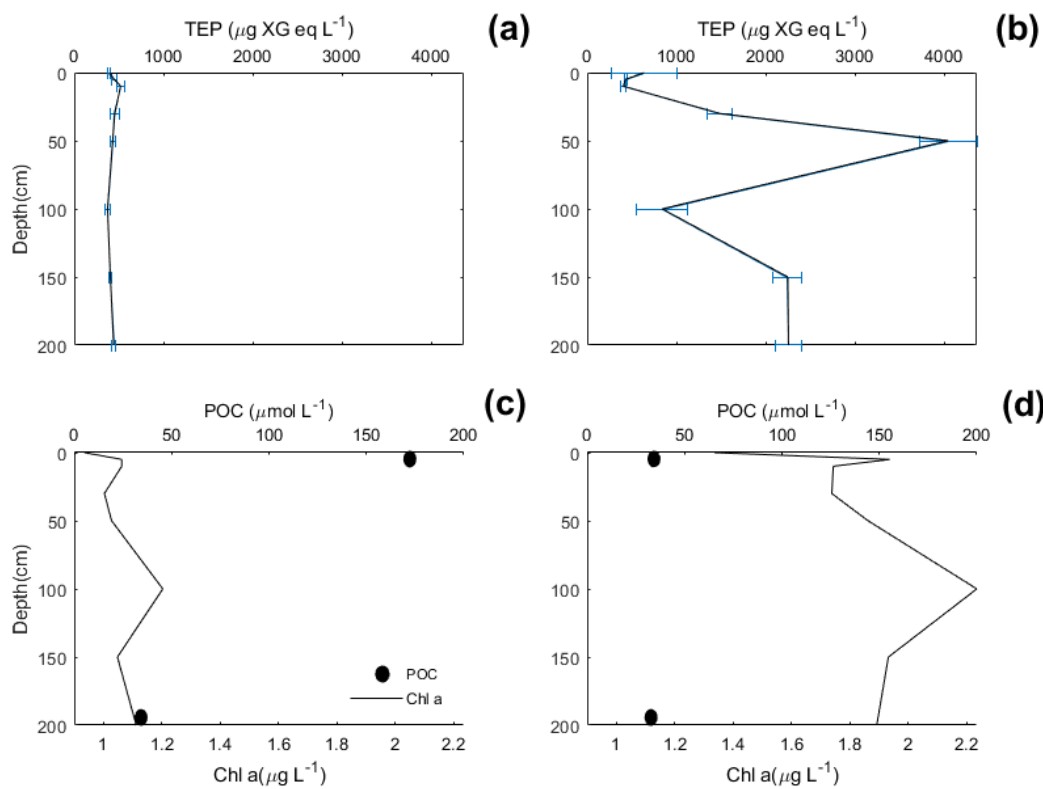

**Figure 3:** Vertical profiles from stations 5 and 9 during the Baltic cruise, showing the vertical distribution of TEP (**a,b**) and Chl *a*/POC (**c,d**). Stations were chosen to represent the general vertical TEP trends seen in the first and second half of the cruise.





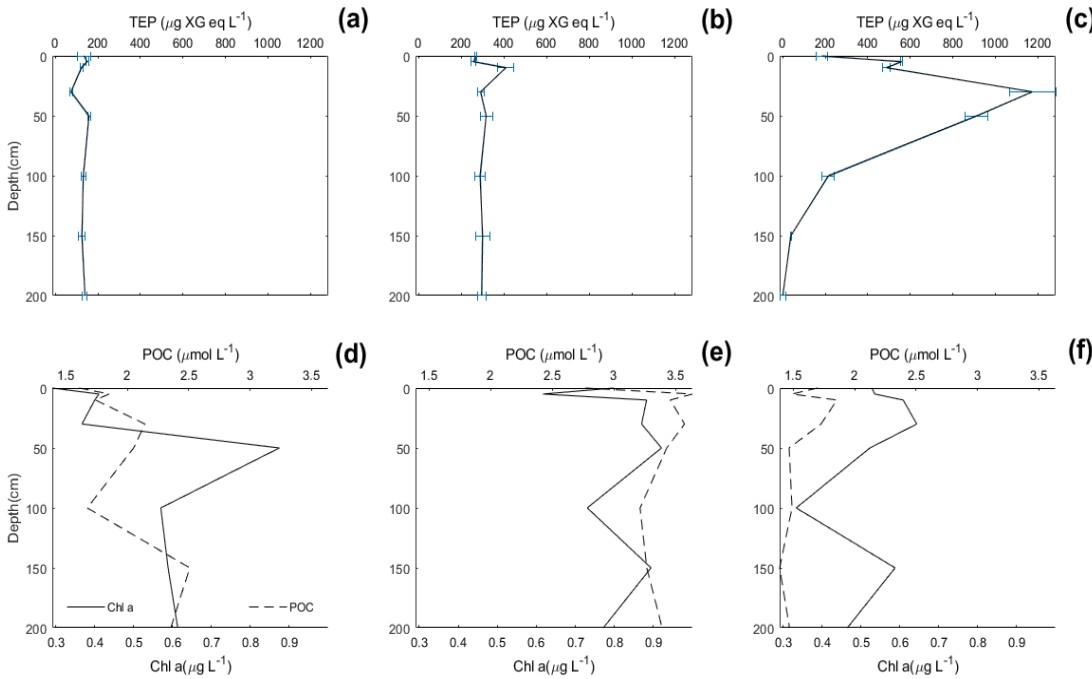

**Figure 4:** Vertical profiles from stations 3, 7, 12 during the Norwegian cruise, showing the vertical distribution of
TEP (**a,b,c**) and Chl *a*/POC (**d,e,f**). Stations were chosen based on the minimum, median, and maximum vertical
variance of TEP.





**Table 1:** S³ data with averages ± SD from 2 hours surrounding discreet sampling and 24hr average for PAR and SI data. HE491 St. 7 was sampled in morning compared to the rest which were sampled in afternoon.

| Campaign | Date | Station | Salinity ULW psu | Salinity SML psu | SST °C | PAR µmol m⁻²s⁻¹ | Solar irradiance Wm²⁻ | Wind Speed ms⁻¹ | S³ sensor 24 hour average PAR µmol m⁻²s⁻¹ | Solar irradiance Wm²⁻ |
|---|---|---|---|---|---|---|---|---|---|---|
| Baltic Sea (EMB184) | 01.06.18 | 3 | n/a | n/a | 14 ± 0.18 | 1644 ± 32 | 637 ± 34 | 3.9 | 1235 ± 378 | 637 ± 56 |
| | 02.06.18 | 4 | n/a | n/a | 16 ± 0.15 | 1555 ± 162 | 620 ± 51 | 3.3 | 1287 ± 331 | 638 ± 121 |
| | 03.06.18 | 5 | n/a | n/a | 16.88 ± 0.5 | 1497 ± 97 | 675 ± 189 | 3.6 | 10868 ± 438 | 496 ± 196 |
| | 06.06.18 | 8 | 8.97 ± 0.04 | 9.06 ± 0.04 | 14.22 ± 0.16 | 1444 ± 361 | 580 ± 226 | 3.6 | 1088 ± 568 | 499 ± 252 |
| | 07.06.18 | 9 | 9.05 ± 0.01 | 9.16 ± 0.02 | 14.03 ± 0.06 | 1477 ± 103 | 452 ± 33 | 4.4 | 1176 ± 230 | 601 ± 79 |
| | 08.06.18 | 10 | 8.8 ± 0.03 | 8.84 ± 0.03 | 10.84 ± 0.06 | 1470 ± 107 | 438 ± 35 | 4.2 | 892 ± 663 | 670 ± 81 |
| | 09.06.18 | 11 | 8.97 ± 0.04 | 8.97 ± 0.04 | 16.54 ± 0.11 | 1310 ± 150 | 695 ± 135 | 3.8 | 1197 ± 267 | 526 ± 130 |
| | 10.06.18 | 12 | 8.61 ± 0.04 | 8.57 ± 0.04 | 16.11 ± 0.03 | 1583 ± 73 | 652 ± 55 | 3.8 | 1411 ± 239 | 652 ± 102 |
| Norwegian Sea (HE491) | 10.07.17 | 3 | 34.83 ± 0.09 | 34.28 ± 0.09 | 13.79 ± 0.09 | 918 ± 97 | 369 ± 83 | 4.2 | 715 ± 313 | 331 ± 114 |
| | 11.07.17 | 4 | 32.18 ± 0.09 | 31.88 ± 0.07 | 14.24 ± 0.08 | 903 ± 46 | 442 ± 154 | 6.5 | 870 ± 299 | 498 ± 152 |
| | 12.07.17 | 5 | 2.72 ± 0.12 | 2.75 ± 0.12 | 15.21 ± 0.15 | 821 ± 84 | 303 ± 60 | 1.8 | 837 ± 420 | 437 ± 222 |
| | 13.07.17 | 6 | 8.72 ± 0.19 | 8.79 ± 0.20 | 14.98 ± 0.08 | 926 ± 143 | 299 ± 111 | 4.3 | 641 ± 414 | 268 ± 192 |
| | 15.07.17 | 7 | 28.37 ± 0.15 | 28.22 ± 0.15 | 13.53 ± 0.07 | 256 ± 164 | 308 ± 167 | 4.8 | 683 ± 446 | 308 ± 168 |
| | 16.07.17 | 8 | 11.22 ± 0.16 | 11.29 ± 0.17 | 15.09 ± 0.05 | 415 ± 21 | 216 ± 176 | 6.1 | 421 ± 340 | 238 ± 170 |
| | 17.07.17 | 9 | 3.75 ± 0.08 | 3.61 ± 0.08 | 14.31 ± 0.08 | 553 ± 134 | 187 ± 50 | 7.4 | 377 ± 202 | 181 ± 72 |
| | 19.07.17 | 10 | 33.95 ± 0.07 | 34.01 ± 0.05 | 13.56 ± 0.03 | 1511 ± 133 | 711 ± 24 | 5.8 | 933 ± 526 | 534 ± 205 |
| | 20.07.17 | 11 | 31.33 ± 0.24 | 30.95 ± 0.26 | 15.49 ± 0.35 | 989 ± 88 | 572 ± 53 | 1.9 | 649 ± 521 | 556 ± 101 |
| | 22.07.17 | 12 | 34.01 ± 0.13 | 33.68 ± 0.05 | 14.21 ± 0.06 | 949 ± 34 | 540 ± 45 | 2.6 | 849 ± 230 | 427 ± 161 |
| | 23.07.17 | 13 | 33.96 ± 0.16 | 33.56 ± 0.23 | 13.68 ± 0.13 | 641 ± 43 | 245 ± 40 | 5.7 | 437 ± 191 | 184 ± 62 |
| | 24.07.17 | 14 | 24.10 ± 0.59 | 23.93 ± 0.67 | 17.07 ± 0.60 | 1552 ± 140 | 661 ± 46 | 1.2 | 1362 ± 250 | 469 ± 260 |
| | 25.07.17 | 15 | 23.74 ± 0.11 | 23.06 ± 0.21 | 18.62 ± 0.33 | 1039 ± 129 | 448 ± 222 | 1.8 | 828 ± 426 | 455 ± 191 |





**Table 2:** Enrichment factors for each station. EF≥1 shows enrichment in the SML.

| Campaign | Date | Station | Enrichment Factor | | | | | | |
| --- | --- | --- | --- | --- | --- | --- | --- | --- | --- |
| | | | Phosphate | Nitrate | Sililcate | Chl *a* | PON | POC | TEP |
| Baltic Sea (EMB184) | 01.06.18 | 3 | 1.0 | 2.0 | 1.0 | 0.7 | n/a | n/a | 2.4 |
| | 02.06.18 | 4 | 1.0 | 0.7 | 1.0 | 0.7 | 1.4 | 1.8 | 1.0 |
| | 03.06.18 | 5 | 1.0 | 1.0 | 1.0 | 0.8 | 2.0 | 5.1 | 1.0 |
| | 06.06.18 | 8 | 1.0 | 1.0 | 1.0 | 0.6 | n/a | n/a | 0.9 |
| | 07.06.18 | 9 | 1.0 | 1.0 | 1.0 | 0.6 | 0.8 | 0.8 | 0.8 |
| | 08.06.18 | 10 | 1.0 | 1.0 | 1.0 | 0.7 | 0.7 | 0.7 | 0.7 |
| | 09.06.18 | 11 | 1.0 | 1.0 | 1.0 | 0.8 | n/a | n/a | 0.6 |
| | 10.06.18 | 12 | 1.0 | 1.0 | 1.0 | 0.6 | n/a | n/a | 0.2 |
| Norwegian Sea (HE491) | 10.07.17 | 3 | 1.1 | n/a | 1.4 | 1.3 | 0.7 | 1.4 | 1.6 |
| | 11.07.17 | 4 | 0.6 | 0.4 | 0.6 | 1.1 | 0.7 | 0.9 | 1.3 |
| | 12.07.17 | 5 | 1.0 | n/a | 1.1 | 1.6 | 0.8 | 0.9 | 0.3 |
| | 13.07.17 | 6 | 1.0 | 6.3 | 1.0 | 0.6 | 0.7 | 0.7 | 0.6 |
| | 15.07.17 | 7 | 1.0 | 2.0 | 1.6 | 0.5 | 0.7 | 0.8 | 0.8 |
| | 16.07.17 | 8 | 0.9 | 20.6 | 1.0 | 1.1 | 1.0 | 1.0 | 1.1 |
| | 17.07.17 | 9 | 1.0 | 1.8 | 1.0 | 0.7 | 0.7 | 0.8 | 0.9 |
| | 19.07.17 | 10 | 1.0 | n/a | 1.3 | 0.8 | 0.9 | 0.9 | 0.8 |
| | 20.07.17 | 11 | 1.1 | 0.5 | 1.0 | 0.9 | 0.9 | 1.2 | 1.0 |
| | 22.07.17 | 12 | 0.9 | n/a | 1.2 | 1.7 | 0.8 | 0.7 | 0.7 |
| | 23.07.17 | 13 | 1.0 | 0.3 | 1.1 | 1.0 | 0.8 | 0.8 | 0.9 |
| | 24.07.17 | 14 | 2.5 | 0.5 | 2.9 | 0.7 | 0.8 | 0.7 | 1.5 |
| | 25.07.17 | 15 | 0.6 | 1.4 | 1.0 | 0.9 | 0.7 | 0.9 | 1.4 |
| Cape Verde | 20.09.17 | 1 | 2.1 | 1.4 | 2.1 | 1.7 | n/a | n/a | 2.6 |
| | 22.09.17 | 2 | n/a | n/a | n/a | 1.0 | n/a | n/a | 3.1 |
| | 25.09.17 | 3 | 2.4 | 2.3 | 1.4 | 1.0 | 1.2 | 2.5 | 3.4 |
| | 26.09.17 | 4 | 1.6 | 0.9 | 1.4 | 0.6 | n/a | n/a | 0.7 |
| | 27.09.17 | 5 | 0.6 | 1.4 | 0.5 | 0.6 | 1.9 | 3.3 | 2.1 |
| | 28.09.17 | 6 | 1.9 | 1.4 | 0.8 | 0.4 | 1.3 | 4.0 | n/a |
| | 02.10.17 | 7 | 1.2 | 0.4 | 0.4 | 0.8 | 0.9 | 1.9 | 1.1 |
| | 03.10.17 | 8 | 1.1 | 1.5 | 0.7 | 1.4 | 1.2 | 1.6 | 1.4 |
| | 04.10.17 | 9 | n/a | n/a | n/a | n/a | n/a | n/a | n/a |
| | 05.10.17 | 10 | n/a | n/a | n/a | n/a | n/a | n/a | n/a |
| | 06.10.17 | 11 | 0.4 | 1.3 | 1.0 | n/a | n/a | n/a | 1.1 |
| | 07.10.17 | 12 | 0.2 | 1.0 | 0.3 | 0.6 | n/a | n/a | 0.8 |



**Table 3:** Vertical distribution of TEP concentrations (µg XG eq L$^{-1}$), variance between TEP concentration at all depths is shown as an indicator for homogeneity (µg XG eq$^2$ L$^{-2}$).

| **Baltic Sea (EMB 184)** | | | | | | | | | | | | | |
|---|---|---|---|---|---|---|---|---|---|---|---|---|---|
| Station | 3 | 4 | 5 | 8 | 9 | 10 | 11 | 12 | | | | | |
| Variance | 3x10$^4$ | 2x10$^0$ | 2x10$^3$ | 1x10$^3$ | 1x10$^6$ | 4x10$^5$ | 1x10$^5$ | 7x10$^5$ | | | | | |
| 0 | 539±339 | 123±23 | 392±15 | 308±14 | 638±369 | 1340±781 | 1317±101 | 490±91 | | | | | |
| 5 | 345±218 | | 449±29 | 380±47 | 429±16 | 750±281 | 2045±135 | 560±76 | | | | | |
| 10 | 307±408 | | 520±39 | 339±12 | 406±25 | 2935±423 | 2254±161 | 536±46 | | | | | |
| 30 | 630±288 | | 455±54 | 376±49 | 1486±138 | 2003±575 | 2274±120 | 1559±145 | | | | | |
| 50 | 642±293 | | 431±27 | 270±3 | 4046±320 | 2508±761 | 2021±164 | 2587±205 | | | | | |
| 100 | 224±206 | 120±16 | 374±24 | 358±2 | 841±283 | 1896±105 | 2313±83 | 2615±159 | | | | | |
| 150 | 228±113 | | 405±18 | 316±6 | 2248±162 | 2136±137 | 2506 | 1454±149 | | | | | |
| 200 | 658±269 | | 443±21 | 341±32 | 2256±150 | 1591±223 | 2619 | 1162±92 | | | | | |
| **Norwegian Sea (HE491)** | | | | | | | | | | | | | |
| Station | 3 | 4 | 5 | 6 | 7 | 8 | 9 | 10 | 11 | 12 | 13 | 14 | 15 |
| Variance | 2x10$^5$ | 2x10$^3$ | 1x10$^3$ | 3x10$^3$ | 6x10$^2$ | 2x10$^3$ | 7x10$^3$ | 2x10$^3$ | 1x10$^3$ | 2x10$^3$ | 2x10$^3$ | 7x10$^3$ | 2x10$^3$ |
| 0 | 185±28 | 224±12 | 50±36 | 142±22 | 135±33 | 268±6 | 148±12 | 194±7 | 168±16 | 291±12 | 244± 0.1 | 427±2 | 273±17 |
| 5 | 558±5 | 179±3 | 95±29 | 235±10 | 150±8 | 258±11 | 374±14 | 282±17 | 200±10 | 270±7 | 196±8 | 192±12 | 211±3 |
| 10 | 489±18 | 166±13 | 69±30 | 245±6 | 123±8 | 409±38 | 211±8 | 263±8 | 177±21 | 254±26 | 330±10 | 145±9 | 204±14 |
| 30 | 1175±109 | 104±14 | 78±40 | 264±1 | 74±7 | 291±15 | 129±14 | 190±15 | 107±27 | 290±20 | 176±1 | 205±14 | 161±28 |
| 50 | 912±55 | 108±9 | 103±23 | 331±32 | 159±7 | 317±29 | 138±22 | 188±15 | 97±10 | | 243±2 | 155±13 | 261±27 |
| 100 | 214±28 | 173±6 | 113±13 | 309±23 | 132±10 | 288±23 | 129±4 | 274±27 | 143±12 | 391±2 | 260±20 | 241±39 | 208±9 |
| 150 | 39±2 | | 115±15 | 308±13 | 124±16 | 301±35 | 140±11 | 196±9 | | 323±7 | 225±28 | 180±28 | 293±0 |
| 200 | 0±14 | | 130±29 | 284±6 | 139±11 | 297±22 | 114±17 | 274±12 | | | 168±9 | 202±28 | 312±9 |






**Table 4:** Regional comparison of TEP, Chl a and POC enrichment and concentrations. * p < 0.05 analysed using ANOVA Tukey statistical test (95% confidence interval). POC data from the Baltic Sea was excluded due to low n value

| | Baltic vs. Norwegian | | | Baltic vs. Cape Verde | | | Norwegian vs. Cape Verde | | |
|---|---|---|---|---|---|---|---|---|---|
| Concentration | n | Mean Diff | SE | n | Mean Diff | SE | n | Mean Diff | SE |
| TEP-SML | 8** | 457.1 | 112.8 | 8*** | 516.5 | 118 | 9 | 59.38 | 109.2 |
| TEP-ULW | 8** | 879 | 242.4 | 8*** | 996.9 | 242.4 | 11 | 117.9 | 222.5 |
| Chl *a*-SML | 8** | 0.4149 | 0.1156 | 8*** | 0.7157 | 0.1209 | 9* | 0.3008 | 0.1118 |
| Chl *a*-ULW | 8*** | 0.8436 | 0.1654 | 8*** | 1.204 | 0.1625 | 11 | 0.3605 | 0.1486 |
| POC-SML | | | | | | | 6 | -25.25 | 26.23 |
| POC-ULW | | | | | | | 9 | 0.498 | 2.927 |
| Enrichment | | | | | | | | | |
| TEP | 8* | -0.04231 | 0.3108 | 8* | -0.8611 | 0.3361 | 9 | -0.8188 | 0.2999 |
| Chl *a* | 8 | -0.3048 | 0.1522 | 8 | -0.2125 | 0.1646 | 9 | 0.09231 | 0.1469 |
| POC | | | | | | | 6** | -1.76 | 0.5003 |

p<0.05*, p<0.01**, p<0.001***