# Peer review of "Depth is Relative: The importance of depth for TEP in the near surface environment"

_Ocean Science, 2019_

## Referee Comment (RC1) · Anonymous Referee #1 · 2 Aug 2019

General comments: Robinson et al. "Depth is Relative: The Importance of Depth on TEP in the Near Surface Environment" reports the vertical distributions of transparent exopolymer particle (TEP) concentration in the upper 2 m depth in the ocean including the surface microlayer (SML). Since the SML is the top of the ocean, their chemical character is very important for the ocean-atmospheric interactions. The authors investigated three oceans in Europe and reported their values. I considered that the data of this study is treasurable, but the manuscript is not organized. The authors must polish the draft. In particular, the "materials and methods" section is poor. I cannot understand how they observed. Also, they cannot find consistent features of TEP concentration in the SML throughout the observations; it was the weakness of this study. To overcome this weakness, detailed descriptions will be necessary as mentioned. I'm

not a native English speaker, but the authors' English grammar is not good.

Specific comments: Title: The authors entitled as "Depth is Relative: The Importance of Depth on TEP in the Near Surface Environment", but I cannot understand what they want to say, and importance of depth was not shown in the manuscript. Please change the title.

L9: In my opinion and the authors described, TEP is not a single substance, it is a generic name; so anyone cannot say they were major or not. We cannot measure carbon of TEP or weight of TEP exactly, in particular in the field experiments.

L13 "study of TEP enrichments". Before here, the authors did not describe "enrichment", and thus very confusing. Please describe what the authors did before describing the results.

L17: I cannot understand the sentence.

L18: I cannot understand what the authors want to describe. A homogenous and heterogeneous profile of TEP concentration was observed in the same profile?

L20: Results. . .has?

L21: Why the authors can conclude the message.

L30: EPS was used only at once in the text.

L51: Is the investigation the purpose of this study? The investigation is the way to accomplish a purpose.

2.1. Please show maps for better understanding of the observations.

2.2 I cannot understand how many days and how many stations did the authors were observed. What "total" indicate? The sampling days or stations numbers whose location was different?

L80: The citation style was wrong. I think the authors use the reference manager such

as Endnote, but please review before submission. Same mistakes were observed in several parts (L108, L127, L129, L277, L292).

L84: To collect 20L of SML water, 2*10ˆ3 m-2 are necessary when assumed the SML is 100 $\mu$m deep. Is it possible and during the sampling, how long did the sampling do? Does it mean that the waters keep their characters during the samplings?

L87: When the authors measured the temperature profiles in this scale, why they didn't discuss the stratification of physical difference between SML and ULW.

L95: Why the authors judge as near "enough"?

L99: Before measuring the POC, did the authors do the acid treatment? The authors described they use acid-washed GF/F but not mentioned after samplings.

L104: Here, the authors described they measured nitrate and phosphate as nutrients, but silicate concentration was reported in the manuscript. If they used GF/F-filtered seawater for the silicate analysis, how did they avoid the contamination of silicate during the filtering processes?

L106: Chlorophyll is not a proper noun.

L120: How many replicates did the authors were collected? In other words, how did they calculate the error bars of TEP concentrations?

L139: "Anova was significant"? Rewrite such as "the difference was significant in ANOVA"

L141: What is "subsurface bulk water"? This is very important because the authors discussed on enrichment factors.

L144: ug may be $\mu$g

L147: Table 1 and 2 should be supplemental materials. Please show them the contour figure and bar plot, respectively.

Table 1: At L90, the parameters were averaged for 2 hours, but in Table 1, PAR was averaged for 24 hours? Which is correct?

L155: Please show the location of the fjord.

L162: Strictly, the authors defined TEP as transparent exopolymer particle"s", and so they cannot use them as the adverb.

L162: As same as Chlorophyll (L106), phosphate and silicate are not proper nouns.

L166: I cannot find the aim of this sentence. Was The Baltic sea observation the time-series observations? If so, please describe as the date.

L176: "Highest" should be "higher"

L188: Enrichment"s"?

L192: In section 2, the observations Cape Verde is described at the top, but here, at the bottom.

L199: "the" samples?

L199: "the lowest concentration" is the area mean value or a value from a sample?

L200: Phosphate level should be phosphate concentration

L210: The authors should refer to Fig. 1

L221: Vertical distributions should be shown as figures.

L224: In my opinion, the homogeneity and heterogeneity should be shown with CV. I cannot understand why the authors select variance.

L230: consumption" "(Busch...

L230: Please divide the results and discussion. These sentences are the discussion.

L240: When the authors want to use the TKE, please briefly describe in the Materials

and Methods section.

L241: I cannot understand the logic after "Thus".

L250: Cite the references.

L252: "The" previous?

L253: has -> have

L262: Did the authors show biochemical processes? What processes did the authors show?

L260: Which paper assumes the homogenous environments including the SML? I know some paper assumes that chemical character is homogenous in the mixed layer, but they usually did not consider the SML.

L264: relation is not wrong, but it is usually focused on the person to person. I think the relationship is better than relation.

L265: I cannot understand this sentence.

L271: Same with L265.

L274: Why EF was related to the concentration? I cannot see the logic.

L287: Sun et al 2017 should put the end of the sentence.

L291: Same with L274.

L291-302: What is the theme of discussion here?

L303: Again, I cannot find any logics between concentration and EF.

L334: "changed daily" The observations were time-series observations? The authors did repeated observation? If not so, why thay can discuss the daily variation?

L347-351: I cannot understand the English here. This sentence was too long.

L352: What indicate they? Biological sources or chemical characteristics?

4.4 The authors did not investigate the flux. I think that the authors cannot judge that TEP in the SML is produced at the SML, or transported from the ULW.

L367: These are not a new idea of the authors in the present study.

L372: Again, the authors cannot estimate the flux.

Table4: Is this correct? For example, in the column of Baltic vs Norweigan, TEP man diffs were only -0.04 and SE was 0.31; however, their difference was significant. While the Chl a concentration was different -0.3, and its SE value was limited to 0.15, however, its difference was "not" significant. Please re-check.

---

## Referee Comment (RC2) · Anonymous Referee #2 · 8 Aug 2019

In this manuscript, the authors present data on Transparent Exopolymer Particles (TEP), Particulate Organic Carbon (POC), Particulate Organic Nitrogen (PON), Particulate Organic Phosphorous (POP), nutrients (phosphate and nitrate), Chlorophyll a, as well as estimated primary production and (to a limited extend) bacterial cell numbers. The respective data was collected during three oceanographic cruises to the Cape Verde region, the Norwegian Sea & fjords and the Baltic Sea. As indicated in the title, the focus of this study is the Near Surface Environment, with data originating from the uppermost 2m of the water column. In particular, the authors state that their purpose was to analyse the vertical distribution of TEP within the upper layers of the ocean (SML aswell as ULW) and examine correlations of TEP concentrations with the biochemical factors named above. As one conclusion of their study, the authors for-

mulate a quest for establishing standard operating procedures of sampling techniques within their research field.

*******************General remarks*******************

In my opinion this manuscript is well structured and the presented data are analysed and discussed in an appropriate way. The study significantly expands the current data basis by providing data from three cruises. Conclusions drawn from the results are fair, meaning that they are in general not too speculative and that additional data sources are provided if they are helpful in supporting a statement. The study aims are clearly outlined and the language is fluent and precise. Overall I only have minor suggestions to improve the manuscript:

*******************Specific remarks*******************

-L16: 'a novel small-scale vertical sampler' : In my opinion it would be a valuable add-on to provide a photograph or sketch of the HSV (e.g. in the supplementary material).

-L231-232: 'total cell numbers (TCN) and small autotroph profiles': Please provide some more information on the flow cytometry part in the material and methods section. How do you define small autotrophs?

-L253: typo/grammar –> replace 'has been found'

-L259-261: 'this study is the first to...' : No it is not. Consider e.g. Zäncker, Cunliff & Engel 2018 Front.Microbiol. Please change this part and consider to incorporate according reference.

-L275: Do you have any additional data to support that the high enrichment factors in the Cape Verde region are not an artifact due to a different sampling strategy in comparison to the other locations? Did you take for instance any technical replicates using the glass plate technique of Cunliffe and Wurl 2014/Harvey and Burzell 1982 and/or syringe samples on the Norwegian and Baltic Sea cruises?

-L306: Please provide some more information on the tank experiment. For instance, you could give a short outline in the supplementary material.

-Figures: The numbering of the figures is not corresponding to the order in which they occur in the text. For example Fig.1 is appearing very late in the text. Please adjust the order of all figures.

---

## Author Comment (AC1) · 7 Oct 2019

Thank you for your comments, as a native english speaker I tried to pay particular attention to my linguistic choices, however using the specific points you have given I will attempt to re-word certain sections and sentences which may be unclear for an international audience. Regarding the materials and methods sections, we are uncertain as to which aspects were unclear for you, we have described the study areas, the sampling techniques, all analyses and statistical procedures. Additionally, the second reviewer found it precise and fluent. I would like to also point out that we did find consistent features of TEP in the SML. Firstly, for all regions there was a general enrichment of TEP in the SML which is in line with earlier studies. Secondly, within and between regions, TEP was found to be enriched in the oligotrophic waters of Cape Verde even

when abundance was low and TEP enrichment features in the Baltic Sea were linked to biological activity. All of these features are described in section 3.2 in the manuscript.

1. Reviewer Comment: L9: In my opinion and the authors described, TEP is not a single substance, it is a generic name; so anyone cannot say they were major or not. We cannot measure carbon of TEP or weight of TEP exactly, in particular in the field experiments.

Author Response: L9- It's unclear what is meant by "TEP is a generic name and thus can't be described as a major source for OM and carbon transport." TEP is a generic name for a substance which can and has been measured, as well the carbon content of TEP has been measured. The following review article may help (Mari et al. 2017).

2. Reviewer Comment: L13 "study of TEP enrichments". Before here, the authors did not describe "enrichment", and thus very confusing. Please describe what the authors did before describing the results.

Author Response: L13- We were not talking about Enrichment Factors which would indeed need to first be explained. We are simply stating enrichment, which is a common word so does not need the definition explained.

3. Reviewer Comment: L17: I cannot understand the sentence.

Author Response: L17- Since comments from other reviewers found the language to be clear, without further explanation of why the sentence is unclear, it is difficult to re-write it in a more comprehensible way.

4. Reviewer Comment: L18: I cannot understand what the authors want to describe. A homogenous and heterogeneous profile of TEP concentration was observed in the same profile?

Author Response: L17-The sentence has been changed to "For two regions with a total of 20 depth profiles, a maximum variance of TEP concentration of 1.39x106 $\mu$g XG eq2 L-2 between depths and a minimum variance of 6x102 $\mu$g XG eq2 L-2 was

found. Showing that the vertical distribution of TEP was both heterogeneous and homogeneous at times." L18– The following was added to line 17 to show total number of depth profiles taken and show that heterogenous and homogenous profiles were seen throughout these 20 total profiles. "For two regions with a total of 20 depth profiles."

5. Reviewer Comment: L20: Results. . .has?

Author Response: L20- Changed to "have"

6. Reviewer Comment: L21-Why the authors can conclude the message.

Author Response: L21- Due to the relationship of phytoplankton as the main source for TEP production and enrichment, when phytoplankton biomass (chl a used as the common proxy) is low, it means that the enrichment of TEP in the SML must come from another source, while bacteria are also known to produce TEP within the SML and could potentially be a source, bacterial cell counts were not enriched in the SML compared to the ULW during this campaign and so with neither phytoplankton or bacteria as a source, enrichment of TEP in the SML most likely comes from abiotic sources, as there are many abiotic factors which are also known to increase upward transport of TEP and self aggregation within the SML. Please refer to lines 303-317 in the manuscript.

7. Reviewer Comment: L30: EPS was used only at once in the text.

Author Response: L30- EPS has been changed to be fully spelled out.

8. Reviewer Comment: L51: Is the investigation the purpose of this study? The investigation is the way to accomplish a purpose.

Author Response: L51- The following paragraph has been changed to clear up any confusion "The purpose of this study was to understand if there are single drivers of TEP verticle distrubition in the upper 2 meters and if these drivers are consistant between regions. To accomplish this, we investigated the abundance and enrichment of TEP between the SML and ULW, in various regions of the ocean and its relation to biochemical factors. A further aim was to determine if 1 meter depth is a good reference

for TEP and other parameters, and how important depth is in sampling within the top 2 meters. We present data from three field campaigns which show the accumulation of TEP in the upper 2 meters and how it relates to water column stratification, primary production and sea surface conditions"

9. Reviewer Comment: 2.1. Please show maps for better understanding of the observations.

Author Response: 2.1- A map of the study areas has been added.

10. Reviewer Comment: 2.2 I cannot understand how many days and how many stations did the authors were observed. What "total" indicate? The sampling days or stations numbers whose location was different?

Author Response: 2.2- There are 19 days between September 18th and October 6th. Of those 19 days, samples were collected on 12 days, on 7 days samples were not collected due to bad weather as noted in the sentence "weather permitting".

11. Reviewer Comment: L80- The citation style was wrong. I think the authors use the reference manager such as Endnote, but please review before submission. Same mistakes were observed in several parts (L108, L127, L129, L277, L292).

Author Response: L80, L108, L127, L277, L292- citation format fixed.

12. Reviewer Comment: L84- To collect 20L of SML water, $2*10Ë Ę3$ m-2 are necessary when assumed the SML is 100 $\mu$m deep. Is it possible and during the sampling, how long did the sampling do? Does it mean that the waters keep their characters during the samplings?

Author Response: L84- Because the Catamaran has 6 glass discs all simultaneoulsy collecting the SML and a rotation speed of 7rpm, it has a collection rate of approximately 20L h-1. (Ribas-Ribas et al. 2017; Shinki et al. 2012). The catamaran will easily cover an area of 20m x 100m within the one hour of collection. Furthermore, since these are an average of what was collected over an hour, it boasts a better rep-

resentation than for example what would be collected over only 10 minutes.

13. Reviewer Comment: L87- When the authors measured the temperature profiles in this scale, why they didn't discuss the stratification of physical difference between SML and ULW.

Author Response: L87- Please note, tempurature was only measured for the SML and ULW at 1 m depth. Thus stratification could not be discussed for the 0-2m depths. Additionally, no relation was found between TEP concentration or EF and tempurature differences between SML and ULW at 1m.

14. Reviewer Comment: L95- Why the authors judge as near "enough"?

Author Response: L95- if a different body of water was sampled then ULW at 1m depth would consitantly show different results from the other ULW depths, since it was taken from the catamaran with the SML. This was the purpose of collecting 1 m depth ULW from the catamaran instead of taking it from the vertical sampler.

15. Reviewer Comment: L99- Before measuring the POC, did the authors do the acid treatment? The authors described they use acid-washed GF/F but not mentioned after samplings.

Author Response: L99- We did not treat POC samples with acid, as PIC was assumed to be negligable as in no indication of diatom booms.

16. Reviewer Comment: L104- Here, the authors described they measured nitrate and phosphate as nutrients, but silicate concentration was reported in the manuscript. If they used GF/F-filtered seawater for the silicate analysis, how did they avoid the contamination of silicate during the filtering processes?

Author Response: L104- Since we don't discuss silicate data, we have removed it from the manuscript.

17. Reviewer Comment: L106- Chlorophyll is not a proper noun.
Author Response: L106- capitalization removed.

18. Reviewer Comment: L120- How many replicates did the authors were collected? In other words, how did they calculate the error bars of TEP concentrations?

Author Response: L120- TEP samples are taken in triplicates. Sentence changed to "TEP was measured by filtering seawater, in triplicates, onto 0.2 $\mu$m. . .."

19. Reviewer Comment: L139- "Anova was significant"? Rewrite such as "the difference was significant in ANOVA"

Author Response: L139- sentence changed to "when the difference was significant in ANOVA"

20. Reviewer Comment: L141- What is "subsurface bulk water"? This is very important because the authors discussed on enrichment factors.

Author Response: L141- sentence changed to "corresponding ULW taken at 1 m depth."

21. Reviewer Comment: L144- ug may be $\mu$g

Author Response: L144- u changed to $\mu$ throughout document.

22. Reviewer Comment: L147- Table 1 and 2 should be supplemental materials. Please show them the contour figure and bar plot, respectively.

Author Response: L147- sentence changed to "General characteristics of parameters and enrichment for all three campaigns is shown in Fig 2 and Fig 3."

23. Reviewer Comment: Table 1- At L90, the parameters were averaged for 2 hours, but in Table 1, PAR was averaged for 24 hours? Which is correct?

Author Response: Table 1- Averages for 2 hours and 24 hours are given in table 1

24. Reviewer Comment: L155- Please show the location of the fjord.

Author Response: L155- It will be shown in the map suggested in previous comment.

25. Reviewer Comment: L162- Strictly, the authors defined TEP as transparent exopolymer particle"s", and so they cannot use them as the adverb.

Author Response: L162- TEP is not used as an adverb, it is not used modify or qualify anything.

26. Reviewer Comment: L162- As same as Chlorophyll (L106), phosphate and silicate are not proper nouns.

Author Response: L162- Capitalizations removed.

27. Reviewer Comment: L166- I cannot find the aim of this sentence. Was The Baltic sea observation the time-series observations? If so, please describe as the date.

Author Response: L166- sentence has been re-written as " TEP enrichment factors were $\geq$1 for the first half of the cruise (St. 3-5) and <1 for the second half of the cruise (St. 8-12)."

28. Reviewer Comment: L176- "Highest" should be "higher"

Author Response: L176- "Higher" is used to compare two things, for describing 3 or more "highest" should be used.

29. Reviewer Comment: L188- Enrichment"s"?

Author Response: L188- In this case "enrichment" not "enrichments" because the words "and" and "both" were used, thus it distinguishes TEP and Chl a as two separate things.

Reviewer Comment: L192- In section 2, the observations Cape Verde is described at the top, but here, at the bottom.

Author Response: L192- Section 2.2 "Sampling: Cape Verde" was moved to after section 2.3, for the sake of fluidity.

30. Reviewer Comment: L199- "the" samples?

Author Response: L199- "the samples" or "samples" are both technically correct, this is more a matter of style.

31. Reviewer Comment: L199- "the lowest concentration" is the area mean value or a value from a sample?

Author Response: L199- It is a regional area mean value

32. Reviewer Comment: L200- Phosphate level should be phosphate concentration

Author Response: L200- sentence change to "phosphate concentrations"

33. Reviewer Comment: L210- The authors should refer to Fig. 1

Author Response: L210- Sentence changed to "Figure 2 shows that. . ."

34. Reviewer Comment: L221- Vertical distributions should be shown as figures.

Author Response: L221- Sample figures (see figures 3 and 4) are shown to reduce unnecessary clutter of images, while the table is used to present all data.

35. Reviewer Comment: L224- In my opinion, the homogeneity and heterogeneity should be shown with CV. I cannot understand why the authors select variance.

Author Response: L224- You are correct CV would be more appropriate for the mathematical comparison of homogeneity, however we found that the resulting units it produced for TEP might be confusing to read and since we are only using CV or Variance to show comparable homogeneity we chose to stay with variance.

Reviewer Comment L230- Please divide the results and discussion. These sentences are the discussion.

Author Response: L230- Paragraph was moved to the discussion. "This is possibly due to the short depth scale or the region, Yamada et al. 2017 found a positive correlation in the Arctic Ocean but this was on a much larger depth scale (0-4000 meters) and no

correlation was found in the Pacific Ocean. Additionally, any in situ production of TEP or consumption by prokaryotes was likely masked by the large increase in phytoplankton abundance during the second half of the cruise."

Reviewer Comment: L240- When the authors want to use the TKE, please briefly describe in the Materials and Methods section.

Author Response: L240- Sentence changed to "Additionally no correlation was found between TEP and turbulent kinectec energy (TKE), measured with an acoustic Doppler velocimeter (data not shown)."

36. Reviewer Comment: L241- I cannot understand the logic after "Thus".

Author Response: L241- Sentence changed to "TEP profiles shown in Figure 4 were chosen based on min, median and max variance and presented as such, since no correlation could be found to any other parameter." Unlike for the Baltic Sea which had a clear trend of phytoplankton biomass increase between the first and second half of that cruise.

37. Reviewer Comment: L250- Cite the references.

Author Response: L250- Mari et al. 2017 reference added.

38. Reviewer Comment: L252- "The" previous?

Author Response: L252- In this case "Previous" not "The previous"

39. Reviewer Comment: L253- has -> have

Author Response: L253- have is correct and already used

40. Reviewer Comment: L262- Did the authors show biochemical processes? What processes did the authors show?

Author Response: L262- the biochemcial processes are the reasons for the results shown, not the results themselves.

41. Reviewer Comment: L260- Which paper assumes the homogenous environments including the SML? I know some paper assumes that chemical character is homogenous in the mixed layer, but they usually did not consider the SML.

Author Response: L260- The papers mentioned in line 253 which this sentence refers to do not measure the SML separately, they measure from 3-5 meter and below, and disregard the near surface as a homogenously mixed area of bulk water, hence our argument that not only does the SML need to be included but profiles to establish the homogeneity of the ULW are needed.

42. Reviewer Comment: L264- relation is not wrong, but it is usually focused on the person to person. I think the relationship is better than relation.

Author Response: L264- relation is also used for things (ex: where is the store in relation to our office?) and relation tends to imply not only that two things are connected but that the effect one thing has on the other while relationship implies more the status of wether two things are connected or not.

43. Reviewer Comment: L265- I cannot understand this sentence.

Author Response: L265- It means that the SD was higher within each region than between regions.

44. Reviewer Comment: L271- Same with L265.

Author Response: L271- Beginning of sentence changedto "This suggests. . .."

45. Reviewer Comment: L274- Why EF was related to the concentration? I cannot see the logic.

Author Response: L274- Similar reason as comment #7

46. Reviewer Comment: L287- Sun et al 2017 should put the end of the sentence.

Author Response: L287- Reference moved to the end of the sentence.

47. Reviewer Comment: L291- Same with L274

Author Response: L291- Similar reason as comment #7

48. Reviewer Comment: L291-302- What is the theme of discussion here?

Author Response: L291-302- The theme is to discuss how Chl a and TEP concentrations are related between the regions, since many studies find TEP concentration and enrichment to be higher in areas with high phytoplankton biomass (Chl a) and productivity (primary production).

49. Reviewer Comment: L303- Again, I cannot find any logics between concentration and EF

Author Response: L303- Please see comment #7 and further details can be found in Wurl et al. 2011b as reffered to in our manuscript.

50. Reviewer Comment: L334- "changed daily" The observations were time-series observations? The authors did repeated observation? If not so, why thay can discuss the daily variation?

Author Response: L334- Sentence changed to "changed from station to station."

51. Reviewer Comment: L347-351- I cannot understand the English here. This sentence was too long.

Author Response: L347-351- comma changed to period ". Suggesting that. . ."

52. Reviewer Comment: L352- What indicate they? Biological sources or chemical characteristics?

Author Response: L352- It refers to biological sources and is grammatically correct.

53. Reviewer Comment: 4.4 The authors did not investigate the flux. I think that the authors cannot judge that TEP in the SML is produced at the SML, or transported from the ULW.

Author Response: 4.4- That is correct that we did not measure fluxes, but this discussion is based on the vertical distribution of TEP, here we want to highlight that TEP can have an upward flux as a particle due to it's boyancy. That is helpful in order to understand the observed vertical distributions of TEP in the upper meters of the oceans.

54. Reviewer Comment: L367- These are not a new idea of the authors in the present study

Author Response: L367- We do not claim any new ideas in line 367 and properly cited references to highlight phytoplankton as a main source of TEP.

55. Reviewer Comment: L372- Again, the authors cannot estimate the flux.

Author Response: L372- replaced "conclude" with "suggest"

56. Reviewer Comment: Table4- Is this correct? For example, in the column of Baltic vs Norweigan, TEP man diffs were only -0.04 and SE was 0.31; however, their difference was significant. While the Chl a concentration was different -0.3, and its SE value was limited to 0.15, however, its difference was "not" significant. Please re-check.

Author Response: Table 4- Thank you for catching this, indeed the wrong numbers were given, all data was checked again and fixed.

Citations Mentioned 57. Mari, X., U. Passow, C. Migon, A. B. Burd, and L. Legendre. 2017. Transparent exopolymer particles: Effects on carbon cycling in the ocean. Progress in Oceanography 151: 13-37.

58. Ribas-Ribas, M., N. I. Hamizah Mustaffa, J. Rahlff, C. Stolle, and O. Wurl. 2017. Sea Surface Scanner (S3): A Catamaran for High-Resolution Measurements of Biogeochemical Properties of the Sea Surface Microlayer. Journal of Atmospheric and Oceanic Technology 34: 1433-1448.

59. Shinki, M., M. Wendeberg, S. Vagle, J. T. Cullen, and D. K. Hore. 2012. Characterization of adsorbed microlayer thickness on an oceanic glass plate sampler. Limnology

and Oceanography: Methods 10: 728-735.

60. Yamada, Y. and others 2017. Transparent exopolymer particles (TEP) in the deep ocean: full-depth distribution patterns and contribution to the organic carbon pool. Marine Ecology Progress Series 583: 81-93.
* * *

---

## Author Comment (AC2) · 7 Oct 2019

Thank you for your comments, we found all your suggestions to be helpful in improving the manuscript.

1. Reviewer Comment: L16- 'a novel small-scale vertical sampler' : In my opinion it would be a valuable addon to provide a photograph or sketch of the HSV (e.g. in the supplementary material).

Author Response: L16- Oliver looking for HSV picture

2. Reviewer Comment: -L231-232- 'total cell numbers (TCN) and small autotroph profiles': Please provide some more information on the flow cytometry part in the material and methods section. How do you define small autotrophs?

Author Response: L231-232- The following details have been added: "The total cell numbers (TCN) of prokaryotic and small autotrophic cells were determined by inCow cytometry following a modiïňAed protocol from Marie et al. (2000). For determination of cell numbers, water samples were fixed with glutaraldehyde (1% final concentration), incubated at room temperature for 1 h, and stored at -18°C until further analysis. Prokaryotic cells were stained with SYBR Green I (2.5 mM final concentration, Molecular Probes, Schwerte, Germany) for 30min in the dark. Samples were measured on a flow cytometer (C6 FlowCytometer, BD Bioscience, fluorescence accuracy of FITC <75; PE <50), and cells were counted according to side-scattered light and emitted green fluorescence. We used 1.0  $\mu$ m beads (Fluoresbrite Multifluorescent, Polysciences) as internal reference to monitor the performance of the device. Their cell counts include heterotrophic and photoautotrophic prokaryotes. Pico and nanoautotrophic cells were counted after addition of red ïňĆuorescent latex beads (Polysciences, Eppelheim, Germany) and were detected by their signature in a plot of red (FL3) vs. orange (FL2) inćuorescence, and red inćuorescence vs. side scatter (SSC). We did not further differentiate between different groups of prokaryotic and eukaryotic autotrophs."

3. Reviewer Comment: -L253- typo/grammar -> replace 'has been found'

Author Response: L253: sentence has been changed to "...TEP has been found..."

4. Reviewer Comment: -L259-261: 'this study is the first to. . .' : No it is not. Consider e.g. Zäncker, Cunliff & Engel 2018 Front.Microbiol. Please change this part and consider to incorporate according reference.

Author Response: L259-261: We refer here not to the enrichment in comparison to a single reference depth but to the vertical distribution in the upper 2 meters, i.e. TEP concentrations at 7 depths. Zancker et al. 2018 measured TEP in the SML relative to 20cm depth, thus reporting typical enrichment similar to earlier studies and not vertical gradients. To our best knowledge, our manuscript reports for the first time vertical

gradients of TEP near the ocean's surface.

5. Reviewer Comment: -L275: Do you have any additional data to support that the high enrichment factors in the Cape Verde region are not an artifact due to a different sampling strategy in comparison to the other locations? Did you take for instance any technical replicates using the glass plate technique of Cunliffe and Wurl 2014/Harvey and Burzell 1982 and/or syringe samples on the Norwegian and Baltic Sea cruises?

Author Response: L275: In this study, we did not further compare sampler with rotating glass disks and the glass plate. However, in the study by Shinki et al. (2012) such comparison were made using a similar catamaran with rotating glass disks, and the collected thickness between catamaran and plate samples were comparable. We added the following text to line 277: "While manual sampling techniques were employed in Cape Verde in comparison to rotating glass disc samples in the other campaigns, earlier comparative studies by Shinki et al. (2012) found both methods to collect similar SML thickness and associated biochemical parameters. Since our catamaran was modelled after Shinki et al. (2012) we are therefore able to compare the results from both versions of the glass plate method." Shinki, M., Wendeberg, M., Vagle, S., Cullen, J. T., & Hore, D. K. (2012). Characterization of adsorbed microlayer thickness on an oceanic glass plate sampler. Limnology and Oceanography: Methods, 10(10), 728-735.

6. Reviewer Comment: L306- Please provide some more information on the tank experiment. For instance, you could give a short outline in the supplementary material.

Author Response: L306- The following has been added "The tank was made of 10-mm polyvinyl chloride (PVC) plates in a size of 120 cm length  $\times$  110 cm width  $\times$  100 cm height. The tank had a volume of 1400 L with a 500 L aerosol chamber on top. Materials in contact with seawater were made from TeïňĆon, including liners for the wall using Teflon bags."

7. Reviewer Comment: Figures- The numbering of the figures is not corresponding to

СЗ

the order in which they occur in the text. For example Fig.1 is appearing very late in the text. Please adjust the order of all figures.

Author Response: Figures- Figure order has been adjusted and now includes the new map as figure 1.

---

## Author Response (AR2)

Dear Dr. Hoppema,

Thank you for your suggestions for the improvement of the manuscript. Since we agreed with all of your suggestions and have made each change you suggested, it does not seem necessary to give a point-by-point response for each.

The only point we would like to address is your suggestion to move the last paragraph of the discussion to the conclusion, because we chose not to head this suggestion. While this paragraph does perhaps have a sense of conclusion in it, that paragraph works together with the previous paragraph to explain the two main effects (biological and chemical) on the upward and downward transport of TEP.

Again, we would like to thank you for your input, and we hope that you find this revised version satisfactory.

Sincerely,

Tiera-Brandy Robinson and co-authors

[revised manuscript text omitted]